# Hepatoprotective Effects of *Ixeris chinensis* on Nonalcoholic Fatty Liver Disease Induced by High-Fat Diet in Mice: An Integrated Gut Microbiota and Metabolomic Analysis

**DOI:** 10.3390/molecules27103148

**Published:** 2022-05-14

**Authors:** Wenjie Jin, Sungbo Cho, Namujila Laxi, Terigele Bao, Lili Dai, Hongzhen Yu, Rigeer Qi, Junqing Zhang, Genna Ba, Minghai Fu

**Affiliations:** 1NMPA Key Laboratory of Quality Control of Traditional Chinese Medicine (Mongolian Medicine), School of Mongolian Medicine, Inner Mongolia Minzu University, Tongliao 028000, China; wenjie@126.com (W.J.); blue0555@hotmail.com (S.C.); namujila@126.com (N.L.); xiaoxiaoto1994@163.com (T.B.); lilidai@126.com (L.D.); jxp11223344@163.com (H.Y.); qirigeer@163.com (R.Q.); 2Key Laboratory of Tropical Translational Medicine of Ministry of Education, Hainan Provincial Key Laboratory for Research and Development of Tropical Herbs, School of Pharmacy, Hainan Medical University, Haikou 571199, China; hy0207002@hainmc.edu.cn

**Keywords:** *Ixeris chinensis* (Thunb.) Nakai, gut microbiota, metabolomic, nonalcoholic fatty liver disease, Mongolian medicine, hepatoprotective mechanism

## Abstract

*Ixeris chinensis* (Thunb.) Nakai (IC) is a folk medicinal herb used in Mongolian medical clinics for the treatment of hepatitis and fatty liver diseases even though its pharmacological mechanism has not been well characterized. This study investigated the hepatoprotective mechanism of IC on mice with nonalcoholic fatty liver disease (NAFLD) by integrating gut microbiota and metabolomic analysis. A high-fat diet (HFD) was used to develop nonalcoholic fatty liver disease, after which the mice were treated with oral IC (0.5, 1.5 and 3.0 g/kg) for 10 weeks. HFD induced NAFLD and the therapeutic effects were characterized by pathological and histological evaluations, and the serum indicators were analyzed by ELISA. The gut microbial and metabolite profiles were studied by 16S rRNA sequencing and untargeted metabolomic analysis, respectively. The results showed that the administration of IC resulted in significant decreases in body weight; liver index; serum biomarkers such as ALT, TG, and LDL-C; and the liver inflammatory factors IL-1β, IL-6, and TNF-α. The 16S rRNA sequencing results showed that administration of IC extract altered both the composition and abundance of the gut microbiota. Untargeted metabolomic analysis of liver samples detected a total of 212 metabolites, of which 128 were differentially expressed between the HFD and IC group. IC was found to significantly alter the levels of metabolites such as L-glutamic acid, pyridoxal, ornithine, L-aspartic acid, D-proline, and N4-acetylaminobutanal, which are involved in the regulation of glutamine and glutamate, Vitamin B6 metabolism, and arginine and proline metabolic pathways. Correlation analysis indicated that the effects of the IC extract on metabolites were associated with alterations in the abundance of Akkermansiaceae, Lachnospiraceae, and Muribaculaceae. Our study revealed that IC has a potential hepatoprotective effect in NAFLD and that its function might be linked to improvements in the composition of gut microbiota and their metabolites.

## 1. Introduction

Nonalcoholic fatty liver disease (NAFLD) is the most common chronic liver disease, affecting one-quarter of the world’s population [1]. It is characterized by an excessive accumulation of fat in the liver in the absence of excessive alcohol consumption, usually defined as <20 g per day for women and <30 g per day for men [2]. The pathological symptoms of NAFLD can be categorized in increasing severity into liver steatosis, nonalcoholic steatohepatitis, fibrotic hepatitis, advanced fibrosis, liver cirrhosis, and hepatocellular carcinoma [3]. NAFLD usually occurs in the context of a metabolic syndrome and is closely associated with obesity, insulin resistance, type 2 diabetes, and dyslipidemia [4]. One of the main causes of the nonalcoholic fatty liver is diet, including both unbalanced intake and an unhealthy diet. Studies have shown that high-fat and high-carbohydrate diets are closely related to nonalcoholic fatty liver [5]. The consumption of these high-energy diets results in low satiety, high caloric intake, and the unfavorable modulation of the gut microbiota, which, in turn, accelerates obesity [6]. There is no specific therapy to treat a nonalcoholic fatty liver. The fundamental guideline for NAFLD is an improvement in diet and living habits to counteract obesity [7].

*Ixeris chinensis* (Thunb.) Nakai (IC), belongs to the Asteraceae family and Ixeris genus, is an edible and officinal perennial herb used in Mongolian medical clinics for the treatment of hepatitis, fatty liver, pneumonia, bronchitis, and diarrhea [8,9]. IC is also used in combination with other medicinal herbs in traditional Mongolian medicine for treating liver and gastrointestinal diseases [10]. Its main chemical components are triterpenoids, steroids, glycosides, volatile oils, flavonoids, and phenolic compounds, among which luteolin, apigenin and β-sitosterol are known to have significant hepatoprotective properties [11,12]. Although IC has anti-inflammatory, antioxidant and hepatoprotective effects [13], there is limited understanding of the effects exerted by IC on NAFLD and its associated mechanism on the gut microbiota and metabolites.

The gut microbiota play an important role in human physiology and is considered to be a key player in the development of liver diseases and lipid metabolism regulation [2]. These microorganisms are affected by the dietary intake and physical activity of the host and changes in their metabolite production may affect the functioning of various organs and, ultimately, the health of the host [14]. For example, the microbiota contribute to carbohydrate digestion [15], bile acid metabolism [16], and vitamin synthesis [17], and it plays vital roles in maintaining liver homeostasis. Imbalances in the microbial constituents of the microbiota can lead to disorders of the enterohepatic circulation that are closely associated with the development of NAFLD [18]. About 70% of the liver blood supply comes from the gut through the portal vein, exposing the liver to metabolites produced by the gut microbes [19]. This relationship is often referred to as the liver–gut axis or liver–microbe axis [20]. Studies have shown that the microbiota are associated with nonalcoholic fatty liver and nonalcoholic steatohepatitis in humans and animals [21,22]. Both the gut microbiota and their metabolites have become key mechanisms for the prevention and treatment of nonalcoholic fatty liver [23]. Thus, the current study examined how IC influences high-fat diet-induced NAFLD in mice by the evaluation of the gut microbiota and their associated metabolic changes, as well as using biochemical and histological analyses. A combination of 16S rRNA sequencing and HPLC-TOF/MS-based metabolomic was used to determine the mechanism by which IC exerts its hepatoprotective effects.

## 2. Results

### 2.1. IC Extract Reduced Body Weight, Liver Index, and Fat Tissue in HFD Diet-Induced NAFLD Mice

To investigate whether IC could prevent HFD diet-induced NAFLD, HFD mice received 0.5, 1.5, or 3 g/kg IC extract by oral gavage for 10 weeks. We found that the body weight, liver index, perirenal fat, and epididymal fat were significantly increased in the HFD groups compared to the control mice (*p* < 0.05). Administration of IC extract resulted in significant decreases in body weight, liver index, perirenal fat, and epididymal fat in comparison with the HFD group (*p* < 0.05) (Figure 1).

### 2.2. Effect of IC Extract on Serum Biochemical Markers

The serum levels of ALT, TG, and LDL-C were significantly elevated in the HFD group compared with the control group (*p* < 0.05). Notably, IC administration significantly reduced the serum levels of ALT, TG, and LDL-C after all three doses in the HFD mice (*p* < 0.05). Additionally, although the serum AST level was not significantly increased in HFD mice, there was a significant reduction in the AST level at the high IC extract dose of IC (*p* < 0.05). Although the HFD induced higher TC levels, these were not significantly reduced by treatment with the IC extract (Table 1).

### 2.3. Effect of IC Extract on Liver Histopathology

To assess morphological changes in liver tissues, the livers of the experimental animals were collected and compared using H&E staining. In the control group, the structure of the liver lobule was complete and the liver cells were arranged radially around the central vein. In the HFD group, the cells showed prominent macro-vesicular steatosis compared with the control mice. The liver sections from the IC group showed mild cellular swelling without steatosis compared with cells from the HFD group; the structure of the hepatic lobule in the IC group was more complete and the degree of steatosis was significantly lower than those in the HFD group (Figure 2).

### 2.4. Effect of IC Extract on Liver IL-6, IL-1β, and TNF-α

Although the levels of IL-6, IL-1β, and TNF-α did not increase significantly in the HFD group compared with the control group, IL-1β levels declined relative to the HFD group (*p* < 0.05) after treatment with IC extract (0.5, 1.5, and 3 g/kg). IC extract at the 0.5- and 1.5-g/kg doses reduced the concentrations of IL-6 (*p* < 0.05), while IC extract at the 3 g/kg dose reduced the TNF-α level (*p* < 0.01) (Table 2).

### 2.5. Gut Microbiota Analysis

The Venn diagram shows the OTUs shared between groups and those that are unique to each group. As seen in the Figure 3A, 585 (31.25%) of the total OTUs were common to all three groups. A further 135 OTUs were shared between the control and HFD group, while 181 OTUs were shared between the HFD and the IC groups. Treatment with the IC extract increased the number of OTUs.

The α-diversity analysis reflects the abundance and diversity of a single sample and includes the Shannon, Observed species, and Chao1 indices. As shown in Figure 3D–F, the Shannon, observed species, and Chao1 indices of the IC group are higher than those in the HFD group, indicating that species diversity is high, and the distribution is even. The β-diversity reflects the presence of significant differences in the microbial communities between the samples. In this study, principal coordinate analysis (PCoA) was used to identify differences in species diversity. The distance of each coordinate point represents the degree of aggregation and dispersion between samples. The closer the distance, the closer the species composition of the sample; the larger the distance, the greater the difference between the samples. As shown in Figure 3B,C, the samples of the control group, HFD group, and the IC group are distinct, indicating that the species structure is quite different, while the distances between the samples within the groups are close, indicating that the community structure is similar.

### 2.6. Analysis of Species Abundance

A species profiling histogram can visually reveal the more abundant species and their relative abundance in the different samples in terms of different taxonomic levels. At the phylum level, each group of samples included mostly Firmicutes, Bacteroidota, Proteobacteria, Verrucomicrobiota, Desulfobacterota, Actinobacteriota, Acidobacteriota, Nitrospirota, and Campilobacterota (Figure 4A).

At the family level, each group of samples included mostly Lactobacillaceae, Erysipelotrichaceae, Lachnospiraceae, Muribaculaceae, Akkermansiaceae, Desulfovibrionacea, Unidentified_Bacteria_Lachnospiraceae, Rikenellaceae, Enterobacteriaceae, unidentified_Bacteria_and Desulfovibrionaceae (Figure 4B). More precisely, the abundance of Lachnospiraceae and Muribculaceae in the IC group was significantly reduced in contrast to the HFD group (*p* < 0.05), while the abundance of Erysipelotrichaceae was significantly increased in the IC group in comparison with the HFD group. There was a significant decrease in the number of Akkermaniaceae in the HFD group compared with the control group (*p* < 0.05), which was reversed by IC treatment (Figure 5). Tax4Fun function prediction found that the most enriched bacterial metabolic pathways related to NAFLD were amino acid metabolism, carbohydrate metabolism, energy metabolism, nucleotide metabolism, metabolism of cofactors and vitamins, glycan biosynthesis and metabolism, membrane transport, replication and repair, translation, and signal transduction (Figure 6).

### 2.7. Untargeted Metabolomic Study

#### 2.7.1. Data Quality and Identification of Metabolites

A total of 212 metabolites were detected from 15 liver tissue samples. Of these, 35% were associated with organic acids and derivatives, 27% with lipid and lipid-like molecules, 25% with organoheterocyclic compounds, 8% with organic oxygen compounds, and 5% with nucleosides, nucleotides, and analogs. The results of the principal component analysis (PCA) showed that there were significant differences between the three groups of samples, indicating significant differences in the metabolic status of the three groups of mice. Moreover, the distances of the QC samples were observed to be extremely close, indicating a high reliability of the sample data (Figure 7A–C).

#### 2.7.2. OPLS-DA Analysis

The OPLS-DA scatter plot (Figure 8A,B) shows that there was a significant separation between the samples of the control vs. HFD group and IC vs. HFD groups. As shown in Figure 8C,D, the corresponding OPLS-DA model was established multiple times (the number of times *n* = 200) to obtain the R2 and Q2 values of the random model. In control vs. HFD and IC vs. HFD, the R2Y values (respectively R2Y = 0.92; R2Y = 0.97) were close to 1, indicating that the established model conforms to the real situation of the sample data, and the Q2 values (respectively Q2Y = 0.73; Q2Y = 0.50) were close to 1, indicating that if a new sample is added to the model, an approximate distribution will be obtained. Thus, the original model has good stability and shows no over-fitting.

#### 2.7.3. Changes in Metabolites and Biological Metabolic Pathways

The analysis of the differential metabolites was carried out by univariate statistical analysis (Student’s *t*-test). Six representative differential metabolites, namely, D-proline, L-aspartic acid, pyridoxal, ornithine, L-glutamic acid, and N4-acetylaminobutanal, that were identified from the key altered pathways are illustrated in Figure 9A–F. The results showed that IC treatment altered the metabolites in the HFD group.

The KEGG database was used to analyze the pathways involved in the regulation of differential metabolites, and to display the metabolic pathways in the form of bubble charts. As shown in Figure 10, in the IC extract vs. HFD group, nine significantly enriched metabolic pathways were identified. In order of impact: biosynthesis of phenylalanine, tyrosine and tryptophan; metabolism of glutamine and glutamate; Vitamin B6 metabolism; β-alanine metabolism; alanine, aspartic acid and glutamate metabolism; arginine and proline metabolism; histidine metabolism; glutathione metabolism; and pantothenate and CoA biosynthesis. The three most significant metabolic pathways associated with the improvement of nonalcoholic fatty liver by treatment with IC extract were glutamine and glutamate metabolism, vitamin B6 metabolism, and arginine and proline metabolism.

### 2.8. Correlation Analysis between Differential Bacteria and Liver Metabolism

Spearman correlation coefficients were used to analyze the relationships between differential gut microbiota and liver metabolites during the IC intervention in HFD mice (Figure 11). Akkermansiaceae were up-regulated by IC intervention and were positively correlated with D-proline, pyridoxal, L-phenylalanine, and L-tyrosine amino acids and vitamins, while Lachnospiraceae, down-regulated by IC, negatively correlated with N4-acetylaminobutanal and ornithine. Muribaculaceae, significantly down-regulated by IC intervention, positively correlated with L-glutamic acid and L-aspartic acid.

## 3. Discussion

This study investigated the hepatoprotective effects of IC extract on mice with nonalcoholic fatty liver induced by a high-fat diet, analyzing body weight and the liver index, fat tissues, serum transaminase, liver inflammatory factors, liver pathology, correlation with gut microbiota and liver metabolites. The HFD contained 2% cholesterol, 7% lard, 8.3% egg yolk, 16.7% sucrose, and 66% basic feed [24] and was fed for 12 weeks to successfully induce a NAFLD mouse model. By 12 weeks, the body weight of the mice in the two groups differed significantly. After treatment with the IC extract, there were significant reductions in body weight, liver index, and perirenal and epididymal adipose tissues compared with those of the HFD group, indicating an improvement in liver fat accumulation. Weight gain can significantly promote liver fat accumulation, and weight loss is the most effective way to remove fat [25]. According to MRI, a 5% reduction in body mass index can reduce liver fat by 25% [26].

NAFLD covers a spectrum of liver conditions the dominant feature of which is abundant hepatic triglyceride (TG) accumulation [27]. NAFLD is strongly related to obesity, diabetes, and dyslipidemia, and is thus considered to be the hepatic manifestation of metabolic syndrome [28]. On the other hand, serum ALT and AST are considered to be markers of liver damage, including fatty liver [29]. Further research has suggested that these two enzymes can be considered as indicators of liver metabolic function [30]. Hence, the levels of serum ALT, AST, TG, TC, HDL-C, and LDL-C were measured in this study. The results showed that there were increases in serum ALT, TG, and LDL-C in the HFD group compared with the control group, indicating abnormal liver metabolic function in the HFD mice. In contrast, the serum ALT, TG, and LDL-C levels were reduced after treatment with the IC extract compared with the HFD group, indicating improved lipid absorption and metabolism. Although the serum AST levels in the HFD group tended to increase, they did not differ significantly from those in the control group, and treatment with IC extract reduced the serum AST level compared with the HFD group. The liver plays an important role in orchestrating glycolipid metabolism [30]. We speculated that IC treatment improved lipid metabolism, reducing lipid accumulation in the liver, thus lowering serum lipid levels. Taken together, these results indicated that IC treatment improves the metabolic function of the liver.

This study also compared pathological changes in the liver appearance and morphology between the groups. The livers in the HFD mice were the color of faded blood and showed marked steatosis compared with those of the control group, confirming the establishment of a nonalcoholic fatty liver model after being fed a high-fat diet for 12 weeks. Compared with the HFD group, the livers in the IC extract group were similar to those of the control group. Furthermore, the livers did not appear fatty, indicating that the IC extract protected against fat accumulation resulting from the high-fat diet-induced nonalcoholic fatty liver. The histological findings of the H&E staining showed that the liver cells in the control group were even in size and neatly arranged. In contrast, the liver cells in the HFD group were swollen with many fat vacuoles of different sizes. Compared with the HFD group, the IC extract groups also showed a certain degree of hepatocellular swelling; however, there were no visible fat vacuoles. Nonalcoholic fatty liver can be classified into nonalcoholic fatty liver (steatosis), nonalcoholic steatohepatitis, liver cirrhosis, and hepatocellular carcinoma according to pathological symptoms. The high-fat diet-induced nonalcoholic fatty liver model usually causes only steatosis and mild inflammation [27]. Therefore, we speculated that the results indicated steatosis of the liver and minor inflammation.

Liver fat accumulation can promote the release of a variety of pro-inflammatory cytokines, such as tumor necrosis factor-α (TNF-α), interleukin-6 (IL-6), interleukin-1β (IL-1β), leading to liver inflammation and cellular damage [31,32]. We observed a tendency towards raised levels of TNF-α, IL-6, and IL-1β in the HFD group although the differences were not significant when compared with the control group. This was consistent with the histopathological results that showed that a high-fat diet induced a nonalcoholic fatty liver model and slight inflammation but not inflammatory nonalcoholic steatohepatitis. Compared with the HFD group, the IC extract groups showed significantly reduced levels of liver IL-6 and IL- 1β, while treatment with 3 g/kg IC extract significantly lowered the IL-1β and TNF-α levels. Previous studies have also reported increases in serum and liver TNF-α levels in patients with nonalcoholic steatohepatitis, with the levels associated with the severity of the histopathology [33]. Similarly, in animal models, serum IL-6 levels were also found to increase in relation to the degree of liver inflammation and fibrosis [34], suggesting that the three IC extract doses have potential inhibitory effects on liver inflammatory factors.

The benefits of medicinal herbs on metabolic disorders have been shown to be associated with changes in the composition of the gut microbiota and their metabolites, which may alter metabolic pathways. The composition of the gut microbiota composition appears to be linked with the health of the host. The microbiome contributes to carbohydrate digestion [15], bile acid metabolism [16], and vitamin [17] and amino acid synthesis [35]. Under normal circumstances, the relationship between the human host and the gut microbiome is mutually beneficial. Accordingly, gut microbiota dysbiosis, defined as an imbalance of the intestinal microbiome, is closely associated with the pathogenesis of NAFLD [36]. In this study, 16S rRNA sequences were used to analyze differences in the composition and abundance of the gut microbiota among the groups. Consistent with previous studies, the α-diversity analysis showed that the Chao1, observed species, and Shannon indices of the control and IC groups were higher than those of the HFD group, indicating that the number and diversity of species in the microbiota were higher in the control and IC groups than in the HFD group. NAFLD patients have been found to have reduced microbiota diversity compared to healthy subjects [37]. The β-diversity analysis showed that the gut microbiota composition differed among groups but was similar within the groups. The OTU analysis showed that IC treatment increased the abundance of the Akkermansiaceae family, while significantly decreasing that of the Lachnospiraceae and Muribaculaceae families. The functional prediction results showed that both carbohydrate and amino acid metabolism were involved in microbiota function. The fecal microbiota revealed that the bacterial composition in mice fed a standard diet was dominated by the Bacteroidaceae and Akkermansiaceae families but was devoid of Lachnospiraceae and Muribaculaceae. In contrast, HFD increased the numbers of bacteria belonging to the Lachnospiraceae and Muribaculaceae families, while reducing both Bacteroidaceae and Akkermansiaceae [38]. Akkermansiaceae have been found to be related to improvements in NAFLD symptoms [19], while Lachnospiraceae are related to the induction of NAFLD [39].

Additionally, the alteration in the microbiome composition and its metabolites directly affect the tight junctions between epithelial cells [40], which influences the reactions to microorganisms and antigens. As described above, the administration of IC modified the hepatic immune response. The gut microbiota influence the immune response and control the stability of the intestine [41]. Thus, it is possible that treatment with IC extract alters the composition of the microbiome which, in turn, reduces the immune response and inflammation caused by NAFLD.

To investigate the mechanisms underlying the effects of IC on NAFLD, we carried out liver metabolomic analysis. It is well-known that the gut–liver axis is a critical link whereby gut microbiota affect liver metabolism. Glutamine and glutamate are important energy substrates that are oxidized by the intestine and immune cells to produce energy, allowing the cells to function optimally and grow [42]. In addition to their role as energy substrates, free glutamine and glutamate are specific precursors of glutathione. Dietary supplementation of glutamine increases glutathione content and lessens oxidative stress in mice fed a high-fat diet [43]. Studies have found that glutamine supplementation is able to protect mice with nonalcoholic steatohepatitis induced by a Western diet [44]. Another study showed that oral glutathione administration for four months in patients with nonalcoholic fatty liver disease reduces both the serum ALT and TG levels and, as glutathione can improve liver metabolism [45]. In this study, it was found that the glutamine and glutamate metabolism pathway had one of the highest impact factors and that L-glutamate is a differential metabolite that regulates this pathway. L-glutamic acid was found to be significantly lowered in the HFD group and increased in the IC group, suggesting that IC regulation of this pathway may account for its protective role against nonalcoholic fatty liver.

Vitamin B6 intake negatively correlated with nonalcoholic fatty liver [46,47]. In this study, the vitamin B6 metabolic pathway was found to be significantly enriched in the IC extract group, indicating that it is activated by IC extract. Furthermore, pyridoxal, a differential metabolite involved in the regulation of vitamin B6 metabolic pathways, was significantly reduced in the HFD group, and increased in the IC extract group, suggesting that IC extract can supplement the content of vitamin B6 in the body, leading to reduced liver fat accumulation and inflammation [48].

Arginine and proline are two functional amino acids that involved in the regulation of lipid metabolism and oxidative stress [49,50,51]. The pathological mechanism underlying NAFLD is complex and influenced by many factors, among which oxidative stress is considered to be the main one leading to liver damage and disease progression [52]. In the current study, many of the differential metabolites are known to be involved in the regulation of this pathway, namely, ornithine, aspartic acid, L-glutamate, dextroproline, and N4-acetaminobutanal. These metabolites were significantly reduced in the HFD model group, while the opposite effect was seen in the IC extract group, demonstrating that the concentrations of these metabolites can be increased and that the arginine and proline metabolic pathways were regulated by the IC extract. Finally, up-regulation of this pathway can reduce obesity and lessen oxidative stress, thus leading to improved lipid metabolism and protection against the development of NAFLD.

## 4. Materials and Methods

### 4.1. Materials

IC was harvested from Horqin grassland, Tongliao, Inner Mongolia, China. The high-fat diet was purchased from Shandong Hengrong Biotechnology Co., Ltd. (No.: SCXK (Lu) 2018 0003). Blood alanine transaminase (ALT), aspartate aminotransferase (AST), triglycerides (TG), total cholesterol (TC), high-density lipoprotein-cholesterol (HDL-C), and low-density lipoprotein-cholesterol (LDL-C) kits were purchased from Shenzhen Icubio Biotechnology Co., Ltd. (Shenzhen, China). ELISA kits for measuring the cytokines IL-6, IL-1β, and TNF-α were purchased from Jiangsu Jingmei Biotechnology Co., Ltd. (Jingmei, China). Hematoxylin and eosin (H&E) stain was purchased from Nanjing Jiancheng Technology Co., Ltd. (Nanjing, China).

### 4.2. Animals and Experimental Design

Six-to-eight-week-old male C57BL/6J mice (initial body weight = 18–20 g) were obtained from Changsheng Biotechnology Co., Ltd. (Liaoning, China). All mice were housed individually in a temperature-controlled room (20–25 °C) with a 14 h-10 h light-dark cycle and allowed seven days for acclimatization with *ad*
*libitum* access to water. Thirty mice were randomly divided into five groups (*n* = 6): control group, HFD group, and IC 0.5, 1.5 and 3.0 g/kg groups. The HFD-induced NAFLD model was developed by feeding a high-fat diet (2% cholesterol, 7% lard, 8.3% egg yolk, 16.7% sucrose, with 66% standard normal diet) for 12 weeks, while the control group was given a standard diet. After modeling, mice in the control and HFD group were given 0.5% Carboxymethylcellulose sodium (CMC-Na, Sigma), while mice in the three IC groups were given 0.5, 1.0 or 1.5 g/kg IC extract in 0.5% CMC-Na suspension by gastric lavage once a day. All mice were treated with listed treatments for 10 weeks. Body weights were recorded weekly. At the end of the experiment, blood was collected from the retro-orbital sinus and centrifuged at 3500 rpm for 10 min at 4 ℃. The supernatant was collected and stored at −80 °C. Fresh fecal samples were collected from the cecum and were frozen in liquid nitrogen and stored at −80 °C for analysis. The livers and fat tissues were surgically removed from each mouse, the wet weights were measured, and the tissues were stored at −80 °C for subsequent experiments. All experiments were performed following protocols reviewed and approved by the Institutional Animal Care and Use Committee, Inner Mongolia Minzu University (approval no. NM-LL-2021-06-15-1).

### 4.3. Serum Biochemical Markers Assay

An automatic biochemical analyzer (Ichem-340, Icubio, Shenzhen, China) was used to measure the levels of ALT, AST, TG, TC, HDL-C, and LDL-C in serum. Sample volumes of 150 µL were used for the tests.

### 4.4. H&E Staining

The left liver lobe tissue were fixed with 4% formalin for at least 48 h, after which the liver tissue was dehydrated using an alcohol gradient, cleared with xylene, embedded in paraffin, and sliced into 5 µm-thick sections. The sections were then deparaffinated, stained with H&E, and examined under a microscope (AXIO Vert.A1, Zeiss).

### 4.5. Liver IL-6, IL-1β, and TNF-α Analysis

Liver homogenates (10%) were prepared by the addition of 100 mg liver tissue to a 0.9% sodium chloride solution at a ratio of 1:9 and homogenized using an electric tissue grinder in an ice bath. The homogenate was then centrifuged at 3500 rpm for 10 min and the levels of IL-6, IL-1β, and TNF-α were measured in the supernatant using enzyme-linked immunosorbent assay kits according to the manufacturer’s instructions. A total of 10 µL samples was used for the test, and the absorbance was measured at 450 nm.

### 4.6. Gut Microbiota Analysis

DNA was extracted from fresh mouse feces and was amplified and purified by PCR. The library was prepared using a TruSeq DNA PCR-Free Library Preparation Kit (Illumina, San Diego, CA, USA). Library quantification was performed by Qubit, and sequencing of the 16S rRNA genes was performed on the Illumina Miseq PE250 platform. Sequences were evaluated on the KEGG and Tax4Fun databases for functional group prediction.

### 4.7. Metabolomic Analysis of Liver Samples

Twenty-five milligrams of the samples were weighed into an EP tube and 50 μL of extract solution (methanol–acetonitrile–water = 2:2:1, with isotopically labelled internal standard mixture) was added. The samples were then homogenized at 35 Hz for 4 min and sonicated for 5 min in an ice-water bath. The homogenization and sonication cycles were repeated three times, after which the samples were incubated for 1 h at −40 °C and centrifuged at 12,000 rpm for 15 min at 4 °C. The supernatant was transferred to a fresh glass vial for analysis. Quality control (QC) samples were prepared by mixing an equal aliquot of the supernatants from all of the samples. The liver samples were analyzed for untargeted metabolite profiles using the LC-MS/MS (Q Exactive Orbitrap, Thermo Fisher Scientific, Waltham, MA, USA). LC-MS/MS analyses were performed using a UHPLC system (Vanquish, Thermo Fisher Scientific) with a UPLC BEH Amide column (2.1 mm × 100 mm, 1.7 μm) coupled to Q Exactive HFX mass spectrometer (Orbitrap MS, Thermo Fisher, Waltham, MA, USA). The mobile phase consisted of 25 mmol/L ammonium acetate and 25 mmol/L ammonium hydroxide in water (pH = 9.75) (A) and acetonitrile (B) with the ratio of 5:95%. The auto-sampler temperature was 4 °C, and the injection volume was 2 μL.

The QE HFX mass spectrometer was used for its ability to acquire MS/MS spectra in information-dependent acquisition (IDA) mode under the control of the acquisition software (Xcalibur, Thermo Fisher). In this mode, the software continuously evaluated the full-scan MS spectrum. The ESI source conditions were set as follows: sheath gas flow rate, 30 Arb; Aux gas flow rate, 25 Arb; capillary temperature, 350 °C; full MS resolution, 120,000; MS/MS resolution, 7500; collision energy, 10/30/60 in the NCE mode; spray voltage, 3.6 kV (positive) or −3.2 kV (negative), respectively.

### 4.8. Statistical Analysis

GraphPad Prism 8.0.2 software was used for the statistical analysis and processing of data. The physiological characteristics of mice were expressed as mean ± standard error (X ± SEM). In the experiments, one-way ANOVA was used for the comparison of multiple groups of data, the LSD test was used for multiple comparisons, and *p* < 0.05 indicated that the difference was statistically significant. In the analysis of intestinal flora, Usearch software was used to perform de-chimerism and cluster analysis of the data to obtain the OTUs (Operational Taxonomic Units) for α-diversity and β-diversity analysis. Linear discriminant analysis (LDA) was used to estimate the communities or species that differed significantly between the samples. XploreMET (Metabo-Profile Biotechnology, Shanghai, China) was used to process the raw metabolomic data, aggregation of the QC samples to establish a reference database, comparison of metabolic signals, correction and normalization of missing values, and identification of metabolites. This was followed by a partial least squares discriminant analysis (PLS-DA) and orthogonal partial least squares discriminant analysis (OPLS-DA). Spearman correlation coefficients were used to determine the correlations between differential flora and differential metabolites.

## 5. Conclusions

The hepatoprotective effects and mechanism of IC extract on nonalcoholic fatty liver disease induced by a high-fat diet in mice were evaluated using pharmacological methods integrated with metabolomic analysis and analysis of the gut microbiota. The results of the pharmacological study showed that IC has the potential to reduce body weight and the levels of serum alanine aminotransferase and liver inflammatory factors, and to improve pathological liver symptoms. The mechanism may involve regulation of the intestinal bacteria belonging to the Akkermansiaceae, Lachnospiraceae and Muribaculaceae families, as well as their associated metabolites D-proline, pyridoxal, N4-acetylaminobutanal and ornithine, which are involved in the regulation of glutamine and glutamate metabolism, vitamin B6 metabolism, and arginine and proline metabolism pathways. Further investigations are required to determine the specific relevance of these microbial compositions of the microbiota and their metabolites in hepatoprotective mechanisms.

## Figures and Tables

**Figure 1 molecules-27-03148-f001:**
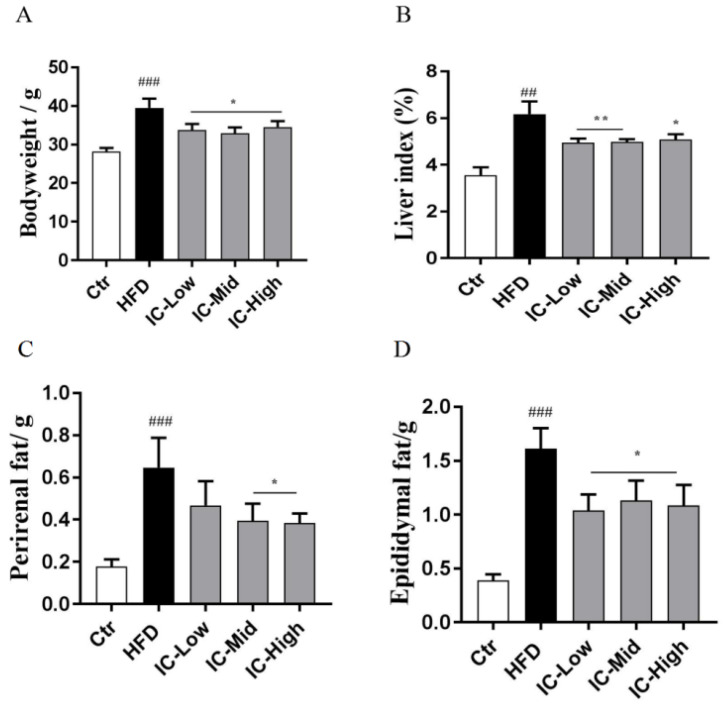
Effects of IC extract on body weight (**A**), liver index (**B**), perirenal fat (**C**), and epididymal fat (**D**). The data represent means ± SEM (*n* = 6). ## *p* < 0.01 and ### *p* < 0.001 versus the control group; * *p* < 0.05 and ** *p* < 0.01 versus the HFD group, respectively. Liver index = (liver weight/body weight) × 100%.

**Figure 2 molecules-27-03148-f002:**
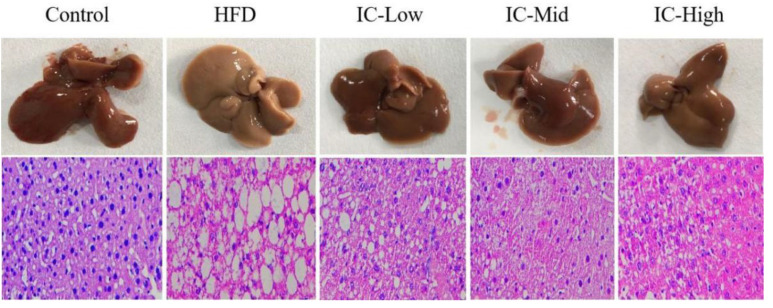
Histopathological changes in liver tissues from the five experimental mouse groups.

**Figure 3 molecules-27-03148-f003:**
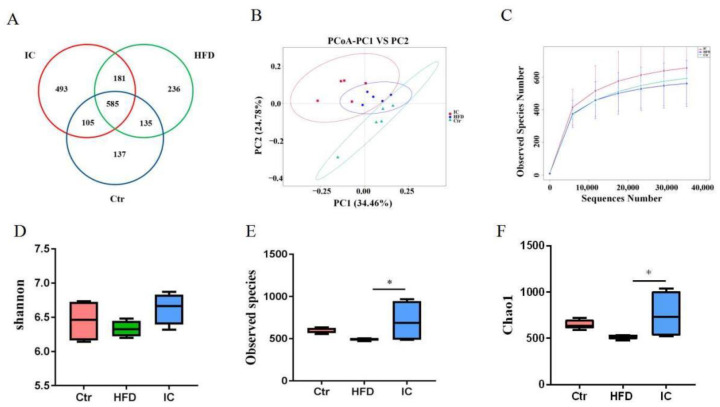
α and β diversity in the fecal samples from the three experimental groups. (**A**). Venn diagram showing the shared and unique OTUs across three experimental groups. (**B**). Principal coordinate analysis (PCoA) ordination plot for the three experimental groups. (**C**). Rarefaction curves showing the average observed species number for three groups. (**D**). Shannon index diversity in the experimental groups. (**E**). Observed species index diversity in the experimental groups. (**F**). Chao1 indices diversity in the experimental groups. * *p* < 0.05, versus the HFD group.

**Figure 4 molecules-27-03148-f004:**
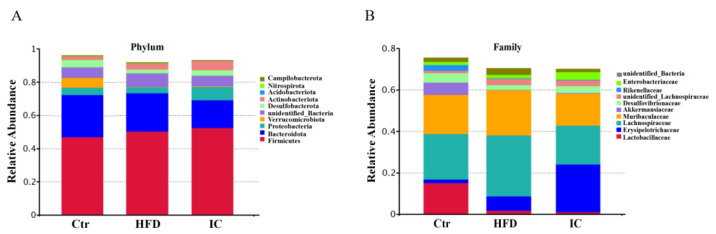
Gut microbiota compositions at the phylum (**A**) and family (**B**) levels for the three experimental groups.

**Figure 5 molecules-27-03148-f005:**
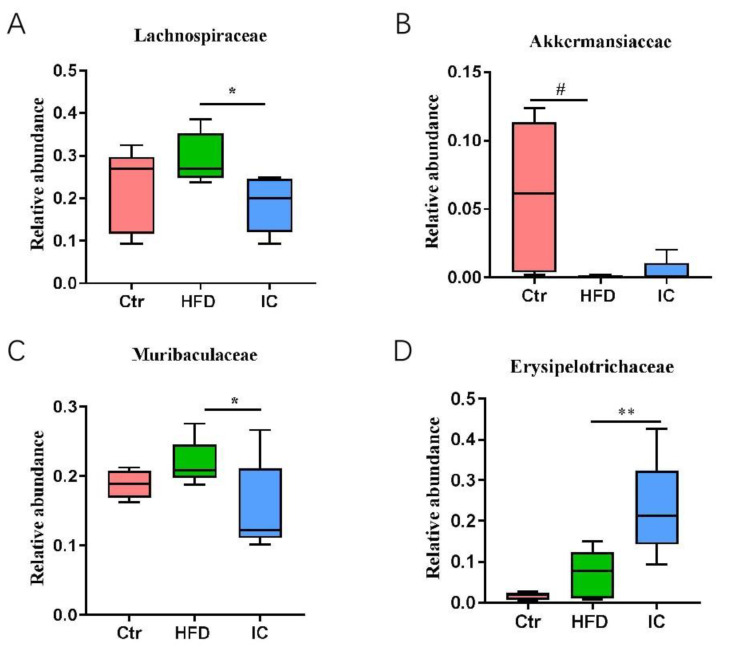
The relative abundance of four representative gut microbiota Lachnospiraceae (**A**), Akkermanisiaccae (**B**), Muribaculaceae (**C**) and Erysipelotrichaceae (**D**) compositions at the Family level. # (*p* < 0.05),versus the control group; ** *p* < 0.01, * *p* < 0.05, versus the HFD group.

**Figure 6 molecules-27-03148-f006:**
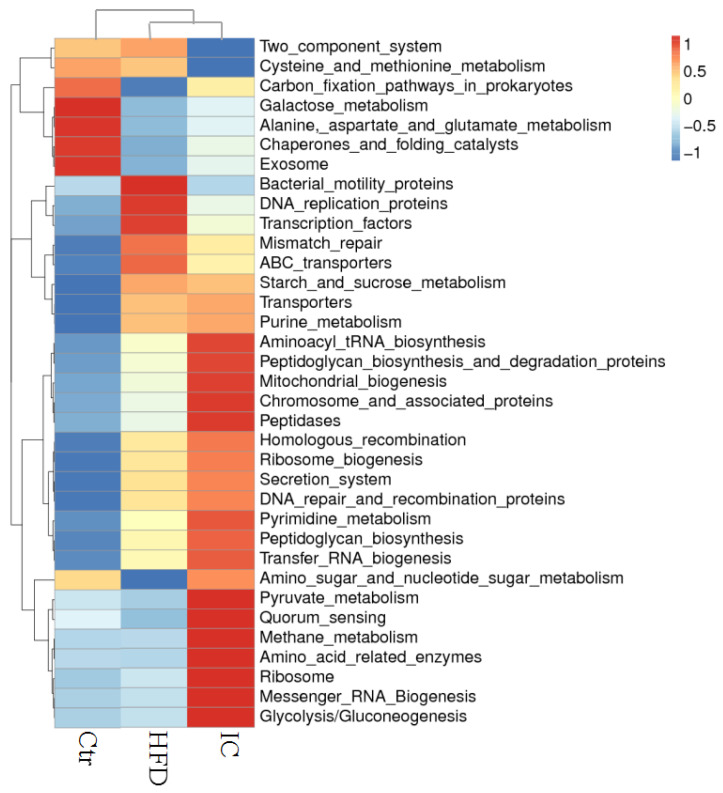
Tax4Fun functional prediction.

**Figure 7 molecules-27-03148-f007:**
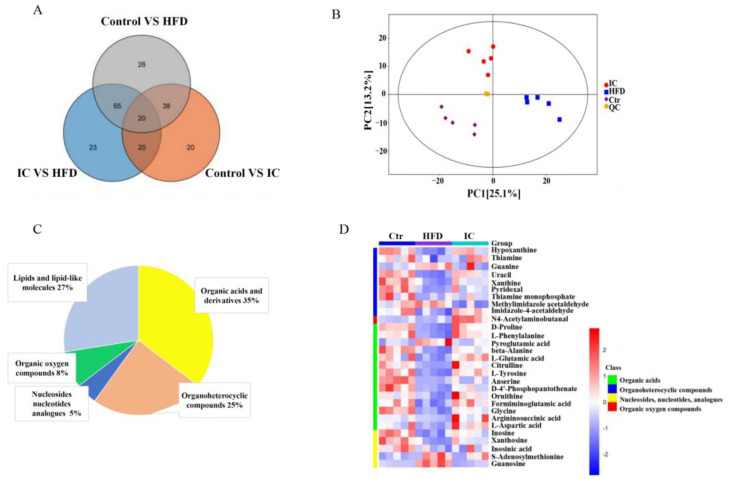
Metabolite classes and compositions in the samples. (**A**). The overall differential metabolites. (**B**). Serum metabolic profiles of the three groups using a principal component analysis (PCA) score plot. (**C**). Pie chart indicating the abundance ratio of different chemical classes of annotated metabolites identified by untargeted metabolic profiling in liver samples. (**D**). Heatmap showing the differences in expression of the differential metabolites in the three groups. Red and blue colors indicate higher or lower expression, respectively, in each sample.

**Figure 8 molecules-27-03148-f008:**
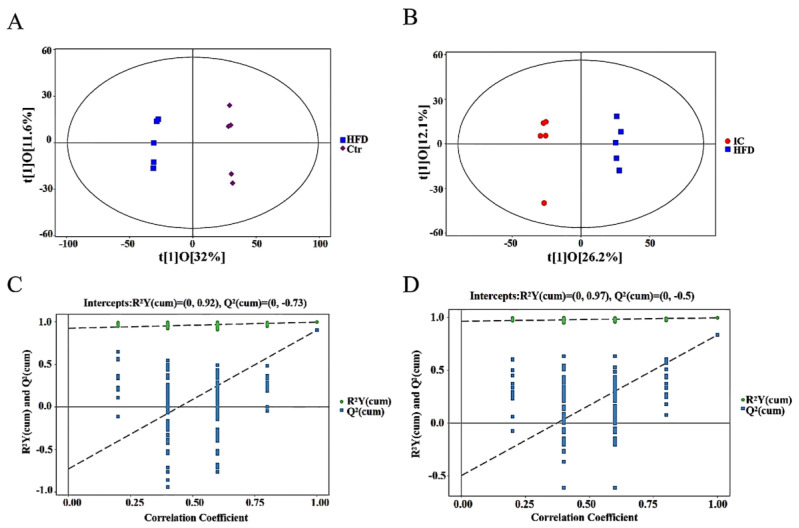
Overall metabolite profile differences between two groups: (**A**). OPLS-DA score scatter plot for Control and HFD groups. (**B**). OPLS-DA score scatter plot for HFD and IC groups. (**C**). OPLS-DA permutation plot for control and HFD groups. (**D**). OPLS-DA permutation plot for HFD and IC groups.

**Figure 9 molecules-27-03148-f009:**
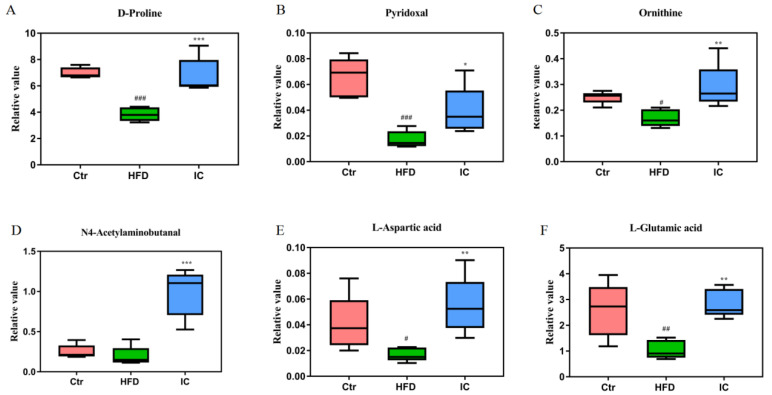
Six representative differentially expressed metabolites D-Proline (**A**), Pyridoxal (**B**), Ornithine (**C**), N4-Acetylaminobutanal (**D**), L-Aspartic acid (**E**) and L-Glutamic acid (**F**) ### *p* < 0.001, ## *p* < 0.01, # *p* < 0.05, versus the control group; *** *p* < 0.01, ** *p* < 0.01, * *p* < 0.05, versus the HFD group.

**Figure 10 molecules-27-03148-f010:**
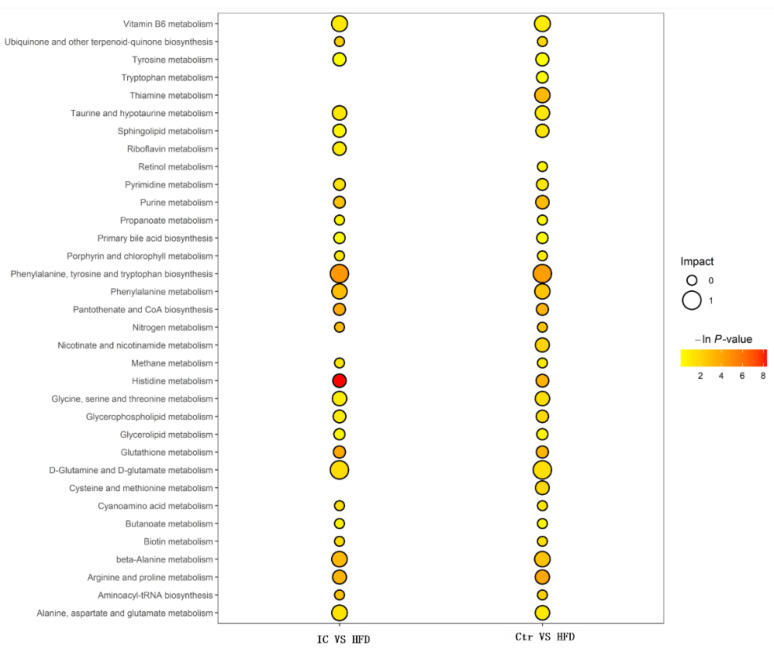
Bubble plot of KEGG pathway analysis.

**Figure 11 molecules-27-03148-f011:**
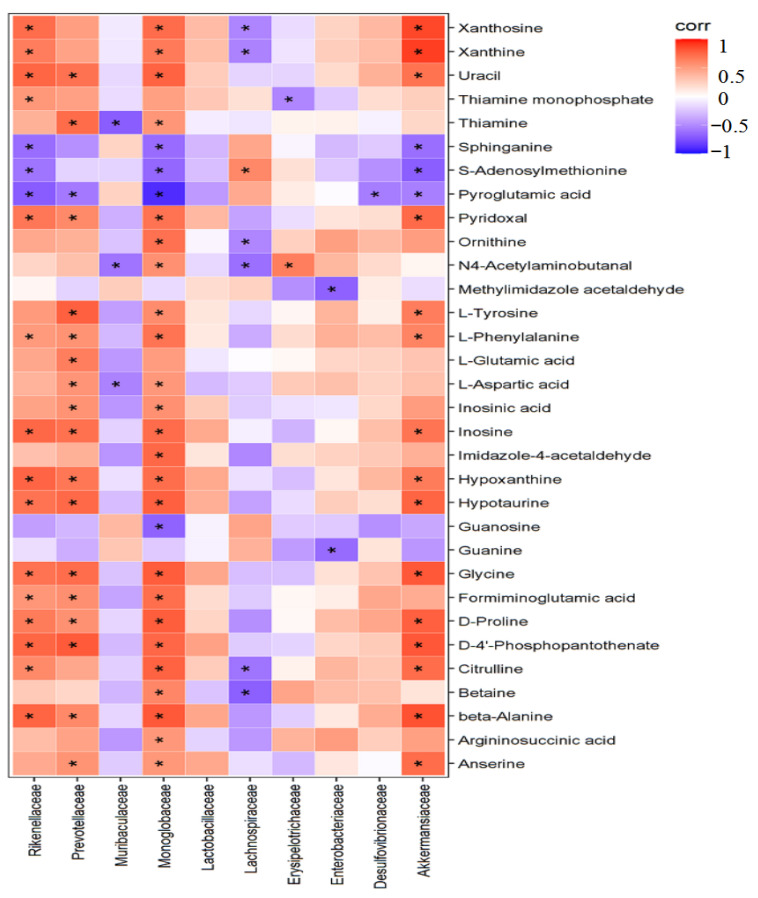
Spearman correlation heatmap of differentially expressed microflora and serum metabolites, * *p* < 0.05.

**Table 1 molecules-27-03148-t001:** Serum biochemical markers in each group (XS, *n* = 6).

Groups	ALT (U/L)	AST (U/L)	TG (mg/dL)	TC (mg/dL)	HDL-C (mmoL/L)	LDL-C (mmoL/L)
Control	31.26 ± 8.65	76.00 ± 13.71	0.32 ± 0.01	2.43 ± 0.13	2.26 ± 0.07	1.18 ± 0.18
HFD	57.13 ± 28.60 #	87.00 ± 16.35	0.38 ± 0.02 #	4.93 ± 1.15 ##	3.79 ± 0.12	3.64 ± 0.68 ##
IC 0.5 g/kg	35.67 ± 7.45 *	88.67 ± 3.67	0.27 ± 0.01 **	3.50 ± 0.50	3.98 ± 0.12	2.56 ± 0.15 *
IC 1.5 g/kg	20.50 ± 10.10 **	82.50 ± 11.29	0.31 ± 0.01 *	3.43 ± 0.29	3.75 ± 0.10	2.46 ± 0.08 *
IC 3.0 g/kg	30.00 ± 19.49 **	75.71 ± 9.69 *	0.30 ± 0.02 *	3.65 ± 0.42	3.80 ± 0.11	2.55 ± 0.10 *

## *p* < 0.01, # *p* < 0.05, versus the control group; ** *p* < 0.01, * *p* < 0.05, versus the HFD group.

**Table 2 molecules-27-03148-t002:** Serum biochemical markers in each group (XS, *n* = 6).

Groups	IL-6 (pg/mL)	IL-1β (pg/mL)	TNF-α (pg/mL)
Control	18.95 ± 0.26	17.91 ± 0.28	106.99 ± 2.91
HFD	19.38 ± 0.34	18.74 ± 0.50	115.60 ± 2.59
IC 0.5 g/kg	17.32 ± 0.97 **	15.80 ± 0.62 **	107.27 ± 2.32
IC 1.5 g/kg	18.33 ± 0.25 *	16.79 ± 0.40 **	111.68 ± 2.59
IC 3.0 g/kg	18.57 ± 0.20	17.42 ± 0.28 *	99.70 ± 6.22 **

** *p* < 0.01, * *p* < 0.05, versus the HFD group.

## Data Availability

Data are contained within this article.

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
