# Peer review of "Hepatoprotective Effects of Ixeris chinensis on Nonalcoholic Fatty Liver Disease Induced by High-Fat Diet in Mice: An Integrated Gut Microbiota and Metabolomic Analysis"

_molecules, 2022, doi:10.3390/molecules27103148_

Round 1

Reviewer 1 Report

The aim of the manuscript written by Jin W. and collaborators was to explore the hepatoprotective mechanism of a folk medicinal herb used in Mongolian medical clinics, i.e. the Ixeris chinensis (Thunb.) Nakai (IC), on mice with nonalcoholic fatty liver disease (NAFLD) induced by high-fat diet. By performing an integrated gut microbiota and untargeted metabolomic analysis, as well as biochemical and histological assays, the authors showed that IC has the potential to ameliorate the pathological pattern typical of the NAFLD possibly acting through the regulation of intestinal bacteria Akkermansiaceae, Lachnospiraceae and Muribaculaceae, as well as their associated metabolites.

I appreciate the authors investigated several aspects of the potential hepatoprotective effects exerted by IC. Furthermore, the manuscript is quite well written. However, as stated by the authors, further investigations are needed to determine the relevance of the identified bacteria and metabolites in the hepatoprotective mechanisms. But this is not my main issue.

I’ve found very unclear the experimental design.

In the abstract is written: “Male C57BL/6J mice were fed with chow or high-fat diet (HFD) with or without IC supplementation”.

In the Methods: “The NAFLD mice were randomly allocated to one of three experimental groups that received IC extract at 0.5 g/kg, 1.5 g/kg, and 3.0 g/kg doses, respectively, by oral gavage for 10 weeks”.

Other that the absolute control group, there was also a chow diet group receiving the IC supplementation? It seems no. How you can exclude that IC can have effects in healthy mice too? Did you perform previous studies? I would briefly explain this in the main text.

Also, as stated by the authors, the NAFLD mice were randomly divided in the 3 groups of supplemented mice which were gavaged for 10 wk. This means that every supplemented group was compared with a NAFLD group (same mice) which was 10 wk old younger?

It wouldn't have been better having a separated NAFLD group in order to compare mice having all the same age?

For this experimental design issues, I prefer the manuscript is considered for major revision, but I’m open to reconsider it depending by the authors answers and by how these issues will be addressed in the main text.

Minor points:

Title: If the authors agree, I think the title would be more appealing in this way “Hepatoprotective effects of Ixeris chinensis on nonalcoholic fatty liver disease induced by high-fat diet in mice. An integrated gut microbiota and metabolomic investigation”.

Also: should Ixeris chinensis be written in italic?

Line 17: metabolomic analysis, not metabolomics analysis.

Line 23: “thus leading to improvement of NAFLD”. I think this statement is too strong since is referred to gut microbiota and further analysis are needed.

Line 24: metabolomic analysis, not metabolomics analysis.

Line 24-25: “of which 29 24 were differentially expressed”. Between which groups?

Line 31: I would state “might be linked”.

Line 56: what Digda-8 and Chaganbong A-13 are? Are commercially available drugs? Online is impossible to find any information and the reference is not helpful.

Line 61: “other pharmacological effects”…please specify at least some of them.

Line 61-63: “there is limited understanding on how IC acts on NAFLD and its associated mechanism on the gut microbiota and metabolites”. Please reformulate as follow: “there is limited understanding on the effects exerted by IC on NAFLD and its associated mechanism on the gut microbiota and metabolites”.

Line 64: I would write “is considered a key player in the development of liver diseases and lipid metabolism regulation”.

Line 65-67: even if I can understand the meaning, the sentence is not clearly written. Please reformulate.

Line 70: “Imbalances in the microbiota”. The imbalance is referred to families or species, right? Please specify.

Line 72: I think there is a space more between “liver” and “blood”.

Line 84-86: please remember to cancel these lines.

Line 91: please explain what liver index represents.

Line 97: How is possible that the n=6 for all the groups if the HFD group was divided in the supplemented ones? Again, there is an explanation lacking in the experimental design part..The same issue is valid for all the other figures and tables.

Line 122: please specify that the increase is not statistically significant.

Line 123-126: please better check all the p value. E.g. in line 124 it is only written p < 0.05 but is also p < 0.01.

Line 133: please specify “in the Figure 3A”

Line 136: please remove Figure 3.

Line 138: please rewrite Chao1, Observed species, and Shannon indices in the same order used after and used in the figures.

Fig. 3 B and C (particularly): please ameliorate the quality and resolution of the images. Also in Fig. 3 D the “s” of Shannon has to be in capital.

Fig. 4: please ameliorate the quality and resolution of the images. Is very difficult to distinguish the phylum and family names. Also it seems to me that the figures should be better disposed throughout the main test.

Line 168: please remove the space before Figure and the comma after 4B.

Line 170: please remove the space before Erysipelotrichaceae.

Line 171: increased not decreased.

Line 180: is not the Fig 5 representing only family level?

Fig. 6: it would be good to find, in all figures, always the same order: Ctr, HFD, IC…In fig. 4 and 7D is also written in different way..

Line 192: remove the space before Figure, and the brackets in A-C.

Fig. 7 and 8: please ameliorate the quality and resolution of the images. Please correct the caption of Fig. 8 (you’re not mentioning C and D).

Line 215: fig. 10 not 11.

Line 291: Change ; with .

Line 302: add a space before the reference.

Line 323: add a space before the reference [35].

Line 418: please modify ad-libitum with ad libitum.

Line 457: remove the space before “water”.

Line 468-469: “The mobile phase consisted of mmol/L ammonium acetate and 25 ammonium hydroxide in water…….” The proportions between (A) and (B) are missing.

Author Response

Response to Reviewer 1 Comments

Overall comments: The aim of the manuscript written by Jin W. and collaborators was to explore the hepatoprotective mechanism of a folk medicinal herb used in Mongolian medical clinics, i.e. the Ixeris chinensis (Thunb.) Nakai (IC), on mice with nonalcoholic fatty liver disease (NAFLD) induced by high-fat diet. By performing an integrated gut microbiota and untargeted metabolomic analysis, as well as biochemical and histological assays, the authors showed that IC has the potential to ameliorate the pathological pattern typical of the NAFLD possibly acting through the regulation of intestinal bacteria Akkermansiaceae, Lachnospiraceae and Muribaculaceae, as well as their associated metabolites.

I appreciate the authors investigated several aspects of the potential hepatoprotective effects exerted by IC. Furthermore, the manuscript is quite well written. However, as stated by the authors, further investigations are needed to determine the relevance of the identified bacteria and metabolites in the hepatoprotective mechanisms. But this is not my main issue.

Response: Thanks very much for the reviewer’s kind and affirmative comments! We appreciate the time and effort that the reviewer dedicated to providing feedback on our manuscript and are grateful for the insightful comments on and valuable improvements to our paper. We are incorporated most suggestions made by the reviewer. Those changes are highlighted within the manuscript as marked in track changes. Please see below, in red, for a point-by-point response to the reviewer’s comments and concerns.

Point 1: I’ve found very unclear the experimental design. In the abstract is written: “Male C57BL/6J mice were fed with chow or high-fat diet (HFD) with or without IC supplementation”. In the Methods: “The NAFLD mice were randomly allocated to one of three experimental groups that received IC extract at 0.5 g/kg, 1.5 g/kg, and 3.0 g/kg doses, respectively, by oral gavage for 10 weeks”.  

Response 1: Thank you for the comments. We edited more clearly in the abstract as below.

Line 17-20: High-fat diet (HFD) was used to develop nonalcoholic fatty liver disease (NAFLD) in mice, which were then treated with oral IC (0.5, 1.5 and 3.0 g/kg) for 10 weeks. HFD induced NAFLD and therapeutic effects were characterized by pathological and histological evaluations, and serum indicators were analyzed by ELISA.

Point 2: Other that the absolute control group, there was also a chow diet group receiving the IC supplementation? It seems no. How you can exclude that IC can have effects in healthy mice too? Did you perform previous studies? I would briefly explain this in the main text.

Response 2: Thank you for the comments. The control mice were fed with a standard diet and, involved in this study to determine the development of NAFLD model group. Thus, the control group did not receive the IC administration. In addition, the study aimed to identify the hepatoprotective effects of IC on HFD-induced NAFLD, thus, the effect of IC supplements on healthy mice were not investigated in the current study.  

Point 3: Also, as stated by the authors, the NAFLD mice were randomly divided in the 3 groups of supplemented mice which were gavaged for 10 wk. This means that every supplemented group was compared with a NAFLD group (same mice) which was 10 wk old younger? It wouldn't have been better having a separated NAFLD group in order to compare mice having all the same age?

Response 3: Sorry for the confusion on the explanation for the experimental design. All mice in this study were same age including control, HFD and three IC-treated groups. NAFLD mouse were developed by HFD for 12weeks, while the control group was given a standard diet. After then, mice in the control and HFD group were given 0.5% Carboxymethylcellulose sodium, while mice in the three IC groups were given 0.5, 1.0 or 1.5 g/kg IC extract in 0.5% CMC-Na suspension by gastric lavage once a day. All mice were treated with listed treatments for 10 weeks. 

Point 4: For this experimental design issues, I prefer the manuscript is considered for major revision, but I’m open to reconsider it depending by the authors answers and by how these issues will be addressed in the main text.

Response 4: Thanks very much for the reviewer’s insightful comments and concern! We have reedited the experimental design more clearly in the Method section as below:

Line 416-424: Thirty mice were randomly divided into five groups (n=6) namely: control group, HFD group, and IC 0.5, 1.5 and 3.0 g/kg groups. The HFD-induced NAFLD model was developed by feeding a high-fat diet (2% cholesterol, 7% lard, 8.3% egg yolk, 16.7 % sucrose, with 66% standard normal diet) for 12 weeks, while the control group was given a standard diet. After modeling, mice in the control and HFD group were given 0.5% Carboxymethylcellulose sodium (CMC-Na, Sigma), while mice in the three IC groups were given 0.5, 1.0 or 1.5 g/kg IC extract in 0.5% CMC-Na suspension by gastric lavage once a day. All mice were treated with listed treatments for 10 weeks.

Minor points:

Point 5: Title: If the authors agree, I think the title would be more appealing in this way “Hepatoprotective effects of Ixeris chinensis on nonalcoholic fatty liver disease induced by high-fat diet in mice. An integrated gut microbiota and metabolomic investigation”.

Also: should Ixeris chinensis be written in italic?

Response 5: Thank you for the suggestion. We agreed to modified the title and the font (Ixeris chinensis) is changed to italic.

Line1: Hepatoprotective effects of Ixeris chinensis on nonalcoholic fatty liver disease induced by high-fat diet in mice. An integrated gut microbiota and metabolomic analysis.

Point 6: Line 17: metabolomic analysis, not metabolomics analysis.

Response 6: Thank you for reviewer’s correction. We have corrected and harmonized all along the main manuscript.

Point 7: Line 23: “thus leading to improvement of NAFLD”. I think this statement is too strong since is referred to gut microbiota and further analysis are needed.

Response 7: Thank you for the comments. We agree the reviewer’s advice and deleted the statement “thus leading to improvement of NAFLD” from the main text. A more speculative statement was added as follows: Our study revealed that IC has a potential hepatoprotective effect in NAFLD and that its function might be linked to improvements in the composition of the gut microbiota and its metabolites (Line 34).

Point 8: Line 24: metabolomic analysis, not metabolomics analysis.

Response 8: Thank you for the correction. We have corrected and harmonized all along the main manuscript.

Point 9: Line 24-25: “of which 29 were differentially expressed”. Between which groups?

Response 9: It was corrected as reviewer’s advice! The number of deferentially expressed metabolites was mistakenly put as 29, after careful check of original data, it was corrected to 128. Thank you!

Line 27: of which 128 were differentially expressed between HFD and IC group.

Point 10: Line 31: I would state “might be linked”.

Response 10: We agree the reviewer’s suggestion and changed as advised. Thank you!

Point 11: Line 56: what Digda-8 and Chaganbong A-13 are? Are commercially available drugs? Online is impossible to find any information and the reference is not helpful.

Response 11: Thanks for the comment. Digda-8 and Chaganbong A-13 are the traditional Mongolian medicine prescriptions that popularly used in the Mongolian medical clinics. To avoid readers’ distraction, we have modified the sentence as below:

Line 57: IC is also used in combination with other medicinal herbs in traditional Mongolian medicine for treating liver and gastrointestinal diseases.

Point 12: Line 61: “other pharmacological effects”…please specify at least some of them.

Response 12: Thanks for the comments! We decided to delete it as we have listed the pharmacological effects of IC in the previous sentences. Thank you!

Point 13: Line 61-63: “there is limited understanding on how IC acts on NAFLD and its associated mechanism on the gut microbiota and metabolites”. Please reformulate as follow: “there is limited understanding on the effects exerted by IC on NAFLD and its associated mechanism on the gut microbiota and metabolites”.

Response 13: Thanks very much for the amendment. We have edited it as advised.

Line 63-65: there is limited understanding on the effects exerted by IC on NAFLD and its associated mechanism on the gut microbiota and metabolites.

Point 14: Line 64: I would write “is considered a key player in the development of liver diseases and lipid metabolism regulation”.

Response 14: Thanks for the suggestion. We have edited it as advised.

Line 66: is considered a key player in the development of liver diseases and lipid metabolism regulation

Point 15: Line 65-67: even if I can understand the meaning, the sentence is not clearly written. Please reformulate.

Response 15: Sorry for the clearance. We re-wrote the sentence as below.

Line 68-70: These microorganisms are affected by the dietary intake and physical activity of the host and changes in their metabolite production may affect the functioning of various organs and, ultimately, the health of the host.

Point 16: Line 70: “Imbalances in the microbiota”. The imbalance is referred to families or species, right? Please specify.

Response 16: Thanks for the comments! We agreed it involves all categories of microbiota and thus we reedited it as below. Thank you!

Line 70: Imbalances in the constituents of the microbiota can lead to disorders of the enterohepatic circulation that are closely associated with the development of NAFLD.

Point 17: Line 72: I think there is a space more between “liver” and “blood”.

Response 17: Thank you. It was deleted.

Point 18: Line 84-86: please remember to cancel these lines.

Response 18: Thank you. We deleted them.

Point 19: Line 91: please explain what liver index represents.

Response 19: Thanks for the advice! Liver index is the ratio of liver of an experimental animal to its bodyweight. It is a commonly used index in toxicology and pharmacological experiment for its simple and sensitive characteristics. Its calculation formulae was added in the main text:

Line 96: Liver index=(liver weight/body weight)×100%.

Point 20: Line 97: How is possible that the n=6 for all the groups if the HFD group was divided in the supplemented ones? Again, there is an explanation lacking in the experimental design part..The same issue is valid for all the other figures and tables.

Response 20: Sorry for the confusion. We developed the model (HFD) groups before IC treatments. And the 24 HFD-induced NAFLD mice were allocated to 4 groups (model and three IC concentrations). We described more in details regarding experiment design (line 416-424).

Point 21: Line 122: please specify that the increase is not statistically significant.

Response 21: Thanks for the suggestion! We have edited the sentence and specified the increase is not statistically significant as below.

Line 120: Although the levels of IL-6, IL-1β, and TNF-α were not increased significantly in the HFD group compared with the control group, after treatment with IC extract (0.5, 1.5, and 3 g/kg), the IL-1β levels declined in comparison with the HFD group (p < 0.05).

Point 22: Line 123-126: please better check all the p value. E.g. in line 124 it is only written p < 0.05 but is also p < 0.01.

Response 22: We have corrected the P values as advised. Thank you!

Point 23: Line 133: please specify “in the Figure 3A”

Response 23: Figure 3A was specified in Line 133. Thank you!

Point 24: Line 136: please remove Figure 3.

Response 24: It was removed as advised, thank you!

Point 25: Line 138: please rewrite Chao1, Observed species, and Shannon indices in the same order used after and used in the figures.

Response 25: We reedited them in the same order used in the figures (line 138). Thank you!

Point 26: Fig. 3 B and C (particularly): please ameliorate the quality and resolution of the images. Also in Fig. 3 D the “s” of Shannon has to be in capital.

Response 26: Thank you for your advice. We have changed them to better quality ones.

Point 27: Fig. 4: please ameliorate the quality and resolution of the images. Is very difficult to distinguish the phylum and family names. Also it seems to me that the figures should be better disposed throughout the main test.

Response 27: Thank you for your suggestion. We replaced the figures in higher resolution.

Point 28: Line 168: please remove the space before Figure and the comma after 4B.

Response 28: Corrected as advised. Thank you!

Point 29: Line 170: please remove the space before Erysipelotrichaceae.

Response 29: Corrected as advised. Thank you!

Point 30: Line 171: increased not decreased.

Response 30: Thank you for your correction. We edited it (line 173).

Point 31: Line 180: is not the Fig 5 representing only family level?

Response 31: Thank you for your correction. The Fig 5 showed the family levels. We reedited the description.

Figure 5. The relative abundance of four representative gut microbiota compositions at the Family level.

Point 32: Fig. 6: it would be good to find, in all figures, always the same order: Ctr, HFD, IC…In fig. 4 and 7D is also written in different way.

Response 32: Thank you for the suggestion. We replaced the figures with same order in Figure 4, Figure 6 and Figure 7D.

Point 33: Line 192: remove the space before Figure, and the brackets in A-C.

Response 33: Corrected as advised. Thank you!

Point 34: Fig. 7 and 8: please ameliorate the quality and resolution of the images. Please correct the caption of Fig. 8 (you’re not mentioning C and D).

Response 34: We replaced the Figure 7 and 8 with higher quality images. The captions of Figure 8 C and D were added as advised. Thank you!

Point 35: Line 215: fig. 10 not 11.

Response 35: Corrected as advised. Thank you!

Point 36: Line 291: Change ; with .

Response 36: Corrected as advised. Thank you!

Point 37: Line 302: add a space before the reference.

Response 37: Corrected as advised. Thank you!

Point 38: Line 323: add a space before the reference [35].

Response 38: Corrected as advised. Thank you!

Point 39: Line 418: please modify ad-libitum with ad libitum.

Response 39: Corrected as advised. Thank you!

Point 40: Line 457: remove the space before “water”.

Response 40: Corrected as advised. Thank you!

Point 41: Line 468-469: “The mobile phase consisted of mmol/L ammonium acetate and 25 ammonium hydroxide in water…….” The proportions between (A) and (B) are missing.

Response 41: Thank you for the comments. The proportion between (A) and (B) is 5:95%.

English was edited by Lexis Academic Editing Service thoroughly. 

Reviewer 2 Report

The authors published a manuscript regarding the hepatoprotective effects of Ixeris chinensis in mice. The manuscript is well writen, showing interesting results on the subject.

General suggestions:

line 84-86: this paragraph must be delected

line 370-402: This is the discussion section and these paragraphs are some kind of revision on the roles of the vitamins/amino acids. These should be reduced or moved to the introduction. 

Author Response

Response to Reviewer 2 Comments 

Overall comments: The authors published a manuscript regarding the hepatoprotective effects of Ixeris chinensis in mice. The manuscript is well written, showing interesting results on the subject.

Response: Thanks very much for the reviewer’s kind and affirmative comments!

General suggestions:

line 84-86: this paragraph must be deleted

Response: Thanks for the suggestion. This paragraph was deleted as advised.

line 370-402: This is the discussion section and these paragraphs are some kind of revision on the roles of the vitamins/amino acids. These should be reduced or moved to the introduction. 

Response: Thanks for the reviewer’s suggestion. We have shorten these two paragraphs and deleted some irrelevant references as advised (see line 380-399).

English was edited by Lexis Academic Editing Service thoroughly. 

Round 2

Reviewer 1 Report

Dear Authors, I really appreciated all the efforts made to ameliorate the quality of the manuscipt which especially needed a better explanation of the experimental design. I think the authors agree with the fact that now is much more clear. Considering all the answers and the improvements made, I'm glad to consider the manuscript accepted for publication.

Best regards.